# Mechanical Strength and Mechanism Analysis of Silt Soil Cured by Straw Ash–Calcium Carbide Slag

**DOI:** 10.3390/ma18020455

**Published:** 2025-01-20

**Authors:** Yue Huang, Wenyuan Xu, Yongcheng Ji, Liang Yang

**Affiliations:** Heilongjiang Provincial Key Laboratory of Road Structure and Green Ecological Technology, Northeast Forestry University, Harbin 150090, China; hy93942326@163.com (Y.H.); yongchengji@126.com (Y.J.); 13144613761@163.com (L.Y.)

**Keywords:** industrial solid waste, solidified silt soil, unconfined compressive strength, stabilized properties, mechanical properties, microstructure

## Abstract

Large-scale engineering projects frequently involve pit excavation and wetland landfill operations, resulting in significant silt accumulation that occupies land and adversely affects the environment. Curing technology offers a solution for reusing this waste silt. In this study, straw ash and calcium carbide slag are proposed as effective curing agents for silt soil. Various indoor tests were conducted to evaluate the mechanical properties of the cured silt soil, while X-ray diffraction (XRD) and scanning electron microscopy (SEM) were used to analyze its mineral composition and micro-morphology. The results showed that increasing the curing agent dosage significantly improved soil strength. Specifically, at a 10% dosage, the California bearing ratio (CBR) value increased to 18.7%, which is 13.4 times higher than untreated silt soil and exceeds road specifications by 8%. At a 20% dosage, the unconfined compressive strength (UCS) value reached 1.38 MPa, meeting the ≥0.8 MPa requirement for roadbeds. Based on economic considerations, a 20% dosage of straw ash–calcium carbide slag was selected as optimal. Microscopic analysis revealed that the addition of these agents promoted the formation of hydrated calcium silicate, filling pores and enhancing the mechanical properties of the cured soil, resulting in a more dense and stable structure.

## 1. Introduction

In recent years, China has witnessed a rising level of urbanization, accompanied by an escalating demand for land resources. During project construction, particularly in the processes of pit excavation and river maintenance, substantial amounts of silt waste are generated. Curing treatment of this waste silt can notably enhance its bearing capacity and facilitate its effective utilization [1]. Previous research and practical tests have revealed that for the curing of silt, common materials primarily encompass inorganic composite materials and certain organic and composite materials, with silicate cement being the most prevalent inorganic material used for this purpose [2]. Silicate cement is obtained by the high-temperature calcination of limestone and clay containing large amounts of oxides, such as SiO_2_, Al_2_O_3_, and Fe_2_O_3_. In the process of mixing with water, the cement will generate hydration products such as calcium silicate hydrate and calcium aluminate hydrate through a hydration reaction. These hydration products can interweave and coagulate the particles in the silt, so that the silt loses mobility and undergoes hardening and agglomeration, thus curing it into a solid material. Although cement curing is a very good choice, the production of cement consumes a lot of non-renewable resources, such as coal, and produces a lot of harmful gases, such as CO_2_ and SO_2_, which have a serious impact on the environment. More importantly, the production cost of cement cannot be ignored. According to market statistics, the cost of silicate cement is CNY 400–600 per ton. Therefore, the method of cement as a traditional curing agent needs to be adjusted from the considerations of resource consumption, environmental pollution, and economic cost [3].

Therefore, the selection of more low-carbon and environmentally friendly curing materials is causing extensive research around the world. Among the different approaches, biomass power plants utilize crop straw and other biomass resources as fuel for power generation, and the straw ash generated after straw combustion often contains heavy metals, SO_2_, nitrogen oxides, and other hazardous substances, which will cause environmental pollution when placed on the land, and it urgently needs to be treated and reused. Domestic scholars such as Liu have explored the effect of different treatment techniques on the activity of biomass ash and the feasibility of biomass ash as a cementitious material. Straw ash, as a by-product originating from the biomass combustion process, is gradually being recognized as having great potential value in the field of building materials, especially as an auxiliary curing material [4]. Calcium carbide slag is the industrial waste generated when calcium carbide reacts with water to produce acetylene gas, and the annual production of calcium carbide slag has been more than 20 million tons in recent years; however, the utilization rate of calcium carbide slag is still less than 50%, and its effective utilization is still a major problem. Based on the scientific concept of curing waste with waste, in this paper, straw ash and calcium carbide slag are used as a new type of curing agent to solidify silt soil and, at the same time, make full use of industrial solid waste to reduce CO_2_ emissions, which is in line with the development requirements of today’s world.

Currently, an increasing number of scholars globally are utilizing diverse waste materials for the investigation of silt stabilization, yielding notable achievements. Zhao et al. found that by comparing the effects of different auxiliary cementitious materials on cement phase composition, the addition of corn stover ash significantly increases the content of C-(A)-S-H in the amorphous cementitious phase during the early stages of specimen development. Furthermore, as the age increases, it also promotes the hydration of C_2_S [5]. Wang et al. conducted experimental research on the stabilization of clayey sands using straw stover ash with lime, cement, and phosphogypsum as exciters. By comparing the exciters, the optimum dosage of straw ash and exciter for different grades of highway subgrade was obtained [6]. Zheng and colleagues employed fibers and phosphogypsum as stabilizing agents to explore the variations in mechanical properties and saturated permeability coefficients with different admixtures of these materials [7]. Fang utilized calcium carbide slag, fly ash, and slag as eco-friendly stabilizing agents, combined with CO_2_ carbonation, to investigate the mechanical properties and underlying mechanisms of silt stabilization [8]. Li and Yang et al. proposed the application of rice husk ash and cement in the silt stabilization process. Their study revealed that rice husk ash significantly enhances the strength of stabilized silt soil and produces hydrated calcium carbonate to fill pores [9]. Zhu used calcium carbide slag and grass ash to formulate a stabilizing agent for the treatment of abandoned soft soil, concluding that these materials undergo a pozzolanic reaction, generating numerous needle-like curing products (C-S-H colloids) and a minor quantity of calcium alumina, thereby improving the compaction of the soft soil structure [10]. Chen’s findings indicated that rice husk ash alone can only marginally improve the compressive strength of soft soil; however, the modification of rice husk ash through the use of carbonated slag and metakaolin as activators significantly enhances the compressive strength, shear properties, and immersion strength of the soil [11]. Basha and Hashim conducted a comprehensive analysis of soft soil compressive strength through compaction tests, unconfined compressive strength tests, California bearing ratio (CBR) tests, X-ray diffraction (XRD) analyses, and scanning electron microscopy (SEM). Their study concluded that both cement and rice husk ash augment soil plasticity, with an optimal admixture of rice husk ash and cement achieving optimal stabilization [12]. These studies collectively demonstrate that certain waste materials can substantially increase soil compressive strength and enhance its stabilization capacity.

## 2. Materials and Methods

Three primary types of materials are utilized in this study: waste silt soil, straw ash derived from a biomass power plant, and corporate emission waste calcium carbide slag. The core experiment encompasses compaction testing, UCS analyses, CBR assessments, XRD analyses, and SEM examinations.

### 2.1. Materials

The test soil was taken from the abandoned soil of a project in Bin County, Harbin City, Heilongjiang Province, and after drying at the construction site, a sealed bucket was selected for handling and storage, as shown in Figure 1b, with the color of gray-black. The test was conducted in accordance with the specification requirements in the “Highway Geotechnical Test Procedure” (JTG 3430-2020) [13], and the basic values of the liquid limit, plastic limit, and maximum dry density of silt soil were obtained [13] as shown in Table 1.

According to Table 1, it can be seen that the natural water content of silt soil is very high, so the selection of curing materials needs to consider materials that have high water absorption capacity and can significantly improve the mechanical properties of silt soil [14].

Straw ash was taken from the solid waste of a biomass power plant. Considering the high content of unburned carbon (LOI up to 15.0%) in the straw ash, a muffle furnace was used to carry out secondary combustion at 400 °C [15]. Figure 2a shows the as-built straw ash before constant temperature firing, which appears pure black in color, and (c) shows the straw ash after 2 h of firing in the muffle furnace (b), which is slightly lightened in color to appear grayish brown.

The calcium carbide slag used in the test was taken from the emission waste of an enterprise in Zhengzhou, Henan Province. As shown in Figure 3, the color of the calcium carbide slag is off-white, and the moisture content is about 20%. The pre-treatment of calcium carbide slag before the formal test, subject to time constraints, does not use natural drying, but uses low-temperature oven drying of the calcium carbide slag for more than 8 h. The main purpose of this is to remove the moisture and impurities in the calcium carbide slag; final, the moisture content of the calcium carbide slag needs to be less than 2%.

In this paper, a laser particle size analyzer produced by Mackick Instruments, Inc. (Tai’an, China) was used to test the silt soil, straw ash, and calcium carbide slag, and the particle gradation of the silt soil, straw ash, and calcium carbide slag was obtained as shown in Figure 4.

X-ray fluorescence is a technology based on physical principles to detect the elements of substances. This paper uses this technology to qualitatively and quantitatively analyze the oxide contents of the silt soil, straw ash, and calcium carbide slag. As shown in the Table 2, the main oxide components of silt soil are SiO_2_ and Al_2_O_3_, and the total proportion of the two is 76.2%. The main oxide of straw ash is SiO_2_, which accounts for about 50%, followed by CaO and Al_2_O_3_, and the oxide of calcium carbide slag is mainly CaO, which accounts for more than 90%.

### 2.2. Methods

In order to determine the optimal ratio of the two materials in the curing agent, in the preliminary test, according to the standardized cementitious sand strength test method to determine the strength of cementitious materials under different ratios of calcium carbide slag and straw ash, it was determined that with a proportion of straw ash and calcium carbide slag (mass ratio) of 6:4, the synergistic reaction between the straw ash and silt soil is the most optimal, and at this time the compressive strength of the silt soil after curing is higher.

#### 2.2.1. Compaction Test

According to the “Highway Geotechnical Test Procedure” (JTG 3430-2020) [13] and “Test Methods of Materials Stabilized with Inorganic Binders for Highway Engineering” (JTG E51-2009) [16], it is necessary to remove the debris in the soil samples and dry them in an oven (the oven temperature is not more than 100 °C). After that, the soil samples were crushed and sieved through a 2 mm sieve, and then the soil materials were mixed using the dry soil method. After that, the soil samples were crushed and sieved through a 2 mm sieve, then mixed by the dry soil method, smothered overnight, and set aside. The next day, a multi-functional electric compactor was used to carry out the compaction test by using the heavy duty compaction method so as to determine the maximum dry density and optimum moisture content of silt soil and cured soil with different dosages of hardener.

#### 2.2.2. UCS Test

According to the “Test Methods of Materials Stabilized with Inorganic Binders for Highway Engineering” (JTG E51-2009), in the requirements of the production of unconfined compressive strength specimens, the specimens for the cylinder must have the following dimensions: a height of 50 mm and a diameter 50 mm. The specimens were compacted by the demolding machine, and the demolded specimens were wrapped in plastic wrap and then sealed in Ziploc bags. Finally, they were placed in a standard curing box. The standard of the curing box is based on the national “Test methods for water requirement of normal consistency, setting time and soundness of the portland cement” (GB 1346-2011) [17]. The temperature must be controlled at 20 ± 1 °C and the relative humidity needs to be ≥95%. When the specimens were cured to the specified age (7, 14, and 28 days), they were taken out from the curing box, and the compressive strength test was carried out using an electronic universal testing machine with the loading speed set to 1 mm/min. Finally, the average of the results of the three parallel specimens was taken as the final result of the specimens, and the unconfined compressive strength of the specimens was calculated.

#### 2.2.3. CBR Test

When the experiment was carried out in accordance with the specifications, a standard indenter was used to press into the soil at a specified rate. The unit pressure at different depths of penetration was recorded and compared with the bearing capacity of the standard crushed stone material, and the value of the CBR was taken as the ratio of the test load applied to the specimen at a penetration of 2.5 mm to that applied to the standard crushed stone material at the same amount of penetration and expressed as a percentage.

#### 2.2.4. Direct Shear Test

The shear strength of soil refers to the ultimate ability of soil to resist shear damage under external loading. The shear stress and the corresponding shear deformation of the soil body are generated by the influence of external loads. Shear damage occurs when the shear stress of the soil reaches the shear strength of the soil. Therefore, it is necessary to study the shear strength of the soil body. A fast shear test of the soil was carried out using a strain-controlled direct shear apparatus. A soil fast shear test refers to the soil body after the application of vertical stress, quickly taking place in 3~5 min using a shear rate of 0.8 mm/min to quickly apply shear stress to make the specimen undergo shear damage.

According to the “Highway Geotechnical Test Procedure” (JTG 3430-2020) [13], firstly, a certain number of specimens were prepared according to the specification requirements, and the size of specimens was 61.8 mm in diameter and 20 mm in height. Each group of specimens was tested according to the four different degrees of stress to determine the cohesion and internal friction angle of the soil under the different contents of calcium carbide slag–straw ash cementitious materials.

#### 2.2.5. Microtesting

The principle of the SEM test is to observe the surface morphology of samples by imaging with secondary electron signals, and in this way, it is applied to observe the morphology and composition of the surface ultrastructure of various solid substances. In this paper, in the SEM using the U.S. Thermo Fisher Scientific-Nova Nao SEM 450 (Waltham, MA, USA), the conservation age of 28 days of curing soil samples was used for research and analysis. The test block with curing soil specimens within the small cut block was used in order to avoid the conductivity of the cut block not being adequate. A flat and smooth section of the surface coated with a thin layer of metal film was taken in order to enhance the electrical conductivity. In this paper, typical specimens (10%, 20%, and 40% of curing agent doping) were taken to observe their microscopic morphology using electron microscope scanning.

The XRD test is based on the principle of an X-ray tube emitted by X-ray irradiation to the specimen to produce diffraction phenomena, with a radiation detector to receive the diffraction line of the X-ray photons amplified by the measurement circuit processing in the display or recording device to give an accurate diffraction line position, intensity, line shape, and other diffraction data.

## 3. Results and Discussion

### 3.1. Solidification Property

The maximum dry density of silt soil was obtained as 1.79 g/cm^3^ and the optimum water content was 14.4% by the heavy duty compaction method. In addition, due to the addition of straw ash and calcium carbide slag, the composition and physical properties of silt soil are bound to change, i.e., the maximum dry density and optimum water content will change. As Figure 5 shows, the relationship between the water content and dry density of silt soil and different curing agent admixtures and the law between the maximum dry density and optimum water content of silt soil and different curing agent admixtures are obtained by analyzing the curves of dry density and water content in the figure.

In Figure 5, the maximum dry density of the silt soil is 1.79/cm^3^; when the dosage of the curing agent is 10%, the maximum dry density of the cured soil decreases to 1.67/cm^3^, and when the dosage of the curing agent reaches 50%, the maximum dry density of the cured soil decreases to 1.43 g/cm^3^. This is because the density of calcium carbide slag and straw ash is lower compared to that of silt soil, so the maximum dry density decreases with the increase in dosage [18]; the more the dosage of straw ash and calcium carbide slag, the more obvious the reaction products between them to fill the pores, so this will lead to a decrease in the dry density with the increase in curing agent. The optimum water content of the cured silt soil increased from 14.4% to 24.5% when the amount of curing agent was increased from 0% to 50%. The reasons for this are as follows: on the one hand, straw ash and calcium carbide slag produce a volcanic ash reaction after being incorporated into silt soil [19], and this reaction process consumes part of the water, thus requiring more water to maintain its stability. On the other hand, during the curing process, calcium carbide slag releases a large amount of calcium ions (Ca+) when it comes into contact with water, and these calcium ions can be exchanged with sodium ions (Na⁺) and potassium ions (K⁺) on the surface of soil particles in silt loam; this exchange action alters the structure of the silt loam particles, which makes the soil particles more likely to aggregate [20]. The curing reaction changes the morphology of the soil particles, making them more irregular or making them form larger agglomerates, and these morphological changes require more water to keep the soil particles stable with each other, so it finally leads to an increase in the optimum water content of the soil after curing.

### 3.2. CBR

The CBR is a test method for evaluating the bearing capacity of subgrade materials, and this method is mainly used to evaluate the bearing capacity of soil base and pavement materials, which is characterized by the ability of the material to resist the deformation of the local load indentation. In this paper, the CBR values of cured soil and silt soil with different curing agent dosages are compared.

Figure 6 shows the trend of the CBR value of the cured silt soil with different straw ash–electrolytic slag dosages. After analyzing the data, as shown in Figure 6, it can be seen that the greater the straw ash–calcium carbide slag dosage, the greater the CBR value; the curing agent dosage of the 10% CBR value gradually and steadily increased until the curing agent dosage of 50%, and the CBR value reached a peak of 28.8%. As the dosage of straw ash and calcium carbide slag increases, the chemical reaction during the curing process consumes more free space, causing changes in the pore structure of the soil [21]. The reduction and refinement of pores help to increase the compactness and strength of the soil. Enhanced inter-particle bonding works as follows: the incorporation of straw ash and calcium carbide slag can increase the contact points and bonding force between soil particles, which makes the soil body better able to transfer and disperse stresses when subjected to stresses, thus improving its bearing capacity, so the CBR value will increase with the increase in the amount of curing agent. The CBR value at the lowest curing agent mixing of 10% was 18.7%, which is 13.4 times that of silt loam’s CBR, and the increase in the CBR value was greatest between 0% and 10%. The increase in the value during this period is the largest, as shown in Table 3, according to the “highway roadbed design specification” (JTGD30-2015) [22] regarding the minimum bearing ratio requirements for roadbed fill, which significantly exceeded the minimum requirements of 8% of the upper roadbed.

### 3.3. Unconfined Compressive Strength

#### 3.3.1. Analysis of UCS Law

The UCS is the compressive strength of a specimen under no lateral pressure, i.e., the ultimate strength of a specimen to resist axial pressure without lateral restriction (zero surrounding pressure). This strength value is usually determined by an unconfined compression test. The UCS at different curing agent dosages and at different maintenance ages is shown in Figure 7.

With the increase in the straw ash–calcium carbide slag dosing, the chemical reaction is also accelerated, which produces more gelling material, so the value of the unconfined compressive strength also increases. Regarding the maintenance time for 7 days, the silt soil UCS is only 0.28 Mpa; when the dosage of curing agent reaches 50%, the compressive strength increases to 1.22 Mpa. When the maintenance time reaches 28 days, the curing agent dosage of the 10% unconfined compressive strength is 0.77 Mpa, and when the dosage of the curing material reaches 50%, the compressive strength increases to 1.22 Mpa and the maintenance time and curing material dosage increases to 1.22 Mpa. The length and the amount of curing material doping have an important effect on the UCS of cured silt soil; when the curing time is increased from 14 d to 28 d, the increase in compressive strength is more obvious than that from 7 d to 14 d. After 28 days of curing, the curing agent doping at 20% increases the compressive strength by 0.61 Mpa compared to that at 10%, which is one of the intervals of the most obvious increase. The increase in strength was attributed to the alkaline environment provided by the calcium carbide slag for the straw ash with volcanic ash activity, in which the reactive SiO_2_ and Al_2_O_3_ in the straw ash were able to undergo a volcanic ash reaction with Ca(OH)_2_ to produce gel products such as hydrated calcium silicate and hydrated calcium aluminate [23]. In addition, the incorporation of straw ash can improve the pore structure and density of hardened soils due to its large specific surface area and high particle fineness. Combined with the test results of the CBR, the CBR value reached the road requirement when the curing agent dosage was 10%, and after that, the larger the curing agent dosage was, the larger the CBR value was; after 28 days of maintenance, the UCS was 0.77 Mpa at 10% (did not satisfy the specification for the high-grade highway lightweight soil subgrade, which needs to be ≥0.8 Mpa (compressive strength requirements), but exceeded 0.8 Mpa at 20%. Considering that the cost of straw ash and calcium carbide slag will be more when more curing agent is added, 20% of curing agent is chosen as the optimal amount under the premise of achieving the basic road performance strength.

#### 3.3.2. Morphological Analysis of Specimens After Damage in UCS Tests

Table 4 shows the morphology of the cured soil before and after destruction of the silt soil and the optimum curing agent dosage at a maintenance age of 28 days.

Under uniaxial compression, the compression of the silt soil specimen was very large at the beginning; then, the specimen gradually appeared from the bottom to the top of the cracks, and when it exceeded its withstand limit, it formed the destruction, which is a typical type of plastic damage.

The solidified soil in the optimal doping showed brittle damage; with compression applied just at the beginning, small cracks appeared on the specimen surface. With the increase in pressure, the cracks quickly extended from the upper part to the lower part, but obvious deformation did not appear on the specimen surface. Then, it suddenly fractured, and the outer layer of the cracked specimen outer peeled off, revealing the inner core. As shown in Figure 8, the remaining inner core continues to bear the load; because of the axial pressure action and obvious shear damage, the inner core is a typical necking type, the morphology of the upper and lower ends are large, and the middle is small.

### 3.4. Straight Shear Test

Figure 9 shows the relationship between soil cohesion and the angle of internal friction with changes in the amount of straw ash–calcium carbide slag in the soil. Figure 9 shows that the cohesion of the soil body increases gradually with the increase in the curing agent dosage. When the dosage of straw ash–calcium carbide dregs is increased from 10% to 50%, the cohesion of the cured soil increases from 59.9 kPa to 91.7 kPa, which is a 55.18% and 137.56% increase in cohesion compared with the untreated soil. The angle of internal friction of the soil also changed with the change in the dosage of straw ash–calcium carbide slag, and the angle of internal friction gradually increased with the increase in the dosage of the curing agent in the soil. The angle of internal friction of the untreated soil was 10.6°, but with 10–50% of the curing agent, the angle of internal friction of the cured soil increased to 24.5–45.2°.

The cohesion of soil is mainly composed of original cohesion, curing cohesion, and capillary cohesion. When the soil is mixed with calcium carbide slag and straw ash and maintained, the chemical components (silicate minerals) in the soil and the active components in the straw ash react chemically with the calcium carbide slag mixed with the soil to form minerals such as calcium silicate (e.g., C-S-H), which have viscoelastic properties. These minerals will be attached to the surface of the clay particles and firmly wrapped around the soil particles so that the curing cohesion of the soil is greatly enhanced, and thus improve the cohesion of the soil. At the same time, the gel attached to the surface of the soil particles and the soil particles will form a larger agglomerated skeleton structure, making the friction roughness of the shear surface greatly increased, thus increasing the angle of internal friction of the soil body.

### 3.5. XRD

Figure 10 shows selected XRD diffraction patterns of the cured silt soil at curing agent contents of 10%, 20%, and 50% at the age of 28 days of maintenance.

From the Figure 10, it can be seen that the straw ash–calcium carbide slag does not change the mineralogical composition of the cured soil significantly, but it has a significant effect on the intensity of the diffraction peaks of the mineralogical components. With the increase in the curing agent dosage from 10% to 50%, the silica diffraction peaks of the straw ash–calcium carbide slag cured soil at 2Θ = 26.6° were gradually weakened, which was due to the fact that the hydration reaction consumed more silica, so the intensity of the silica diffraction peaks was gradually weakened. In addition, the intensity of the calcium hydroxide diffraction peak gradually decreased with the gradual increase in the dosage of straw ash–calcium carbide slag, and the diffraction peak of calcium hydroxide could not be clearly observed even at the curing agent dosage of 50%. This is due to the volcanic ash reaction between silica, the main component of straw ash, and calcium hydroxide, which generates hydrated calcium silicate (C-S-H) that does not melt in water, which makes the intensity of the diffraction peaks of silica and calcium hydroxide decrease. This is consistent with Li’s study that rice husk ash–cement at optimal dosing and at a curing age of 28 days produces reaction products such as hydrated calcium silicate that can increase the strength of cured samples.

### 3.6. SEM

Figure 11 shows an SEM photograph of unconsolidated silt soil magnified 3000 times after 28 days of maintenance. Figure 12 is a photograph of the cured silt soil at a 20% curing agent dosage for 28 days under the SEM magnified 6000 times.

Figure 11 shows that the silt soil has different particle morphologies, mainly some flaky, irregular, sticky grains combined together, with a wide range of particle size distribution, and the surface of some of the particles is rough, with obvious edges and wear traces. In most areas, the particles existed in the form of close stacking, forming a dense microstructure, while in other areas, there were large voids between the particles, forming a loose pore structure. From Figure 12, it can be seen that the micro-morphology of the soil changed significantly after curing, showing a straw ash and calcium carbide slag and silt soil hydration reaction. The formation of the main gelling products of calcium hydrated silicate (C-S-H) on the surface of the silt soil formed a layer of dense cover, with the soil particles and the gelation product combining layer by layer. The layers of soil particles and cementation products become denser, and the pore space is obviously reduced, changing its original surface morphology, which not only improves the compressive strength and seepage resistance of the silt soil but also enhances its resistance to the external environment.

## 4. Conclusions

At present, there is a scarcity of research on the mechanical properties and mechanisms of silt soil stabilized with straw ash–calcium carbide slag both domestically and internationally. In light of this, the present study selects straw ash and calcium carbide slag as stabilizing agents, blends them with silt soil, and examines the mechanical strength characteristics and micro-morphological alterations of the soil through a series of tests, including compaction tests, unconfined compressive strength tests, CBR tests, XRD analyses, and SEM examinations. And the optimum dosage of the curing agent was obtained as 20%, which lays a certain foundation for future experimental research and practical engineering.

Straw ash and calcium carbide slag can significantly improve the strength of cured silt soil. Through a series of indoor mechanical tests and various microscopic tests for theoretical analysis, the mechanism of silt soil cured with straw ash–calcium carbide slag was proven, and the mechanical and microscopic properties of the cured silt soil were analyzed.The plastic limit of cured soil increases with the increase in curing agent mixing, and the plasticity index decreases gradually with the increase in curing agent mixing; with the increase in curing agent, the dry density of cured soil decreases gradually, and the optimum water content increases gradually. The above law is because of the hydration reaction and the exchange between ions.When the dosage of the curing agent is 10%, the CBR value reaches 18.7% and meets the requirement of 8% of roadbed filler; then, the more the dosage of the curing agent is, the more the CBR value of the cured soil increases, which proves that the cured silt soil has good road performance.Twenty-eight days after the maintenance of cured soil, the unconfined compressive strength increased significantly and reached 1.38 Mpa at 20% of curing agent doping, which meets the requirement of the unconfined compressive strength of a basic roadbed being ≥0.8 Mpa. According to the most economical approach, 20% of curing agent doping was selected as the optimal doping.The addition of the straw ash–calcium carbide slag curing agent can promote the formation of hydration products. The curing process resulted in the formation of hydrated calcium silicate and other gel products that have excellent bonding properties and stability, thus giving the cured soil good mechanical properties and environmental stability.

## Figures and Tables

**Figure 1 materials-18-00455-f001:**
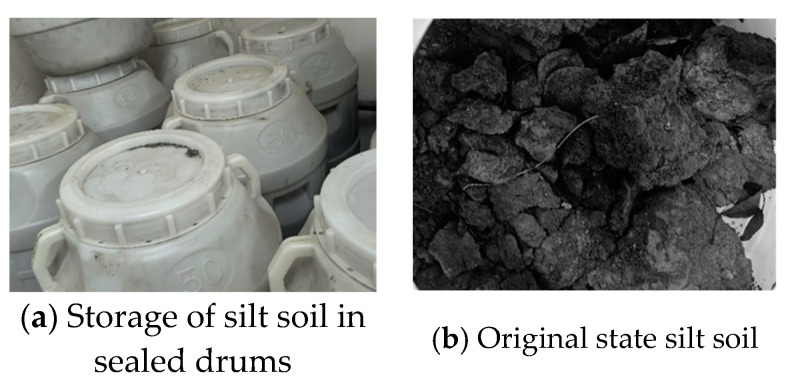
Silt soil.

**Figure 2 materials-18-00455-f002:**
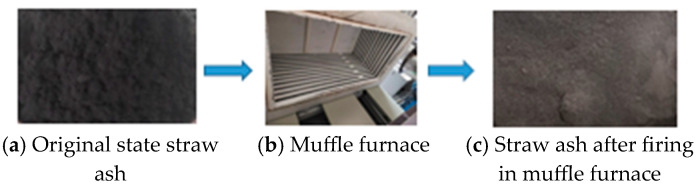
Treatment process of straw ash by muffle furnace.(incineration at 400 °C for 2 h).

**Figure 3 materials-18-00455-f003:**
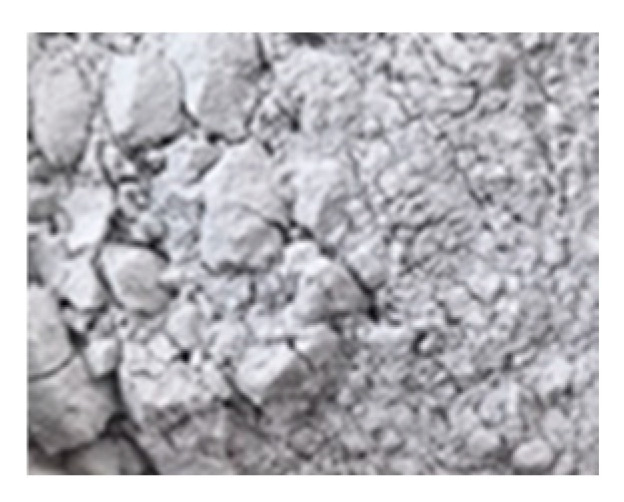
Calcium carbide slag.

**Figure 4 materials-18-00455-f004:**
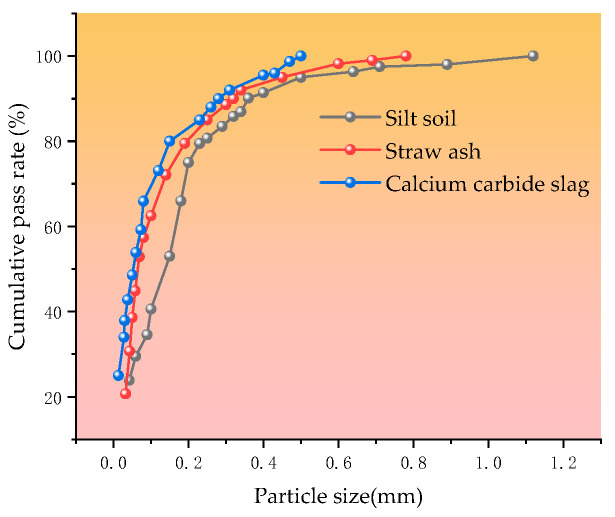
Grading curves for silt soil, straw ash, and calcium carbide slag.

**Figure 5 materials-18-00455-f005:**
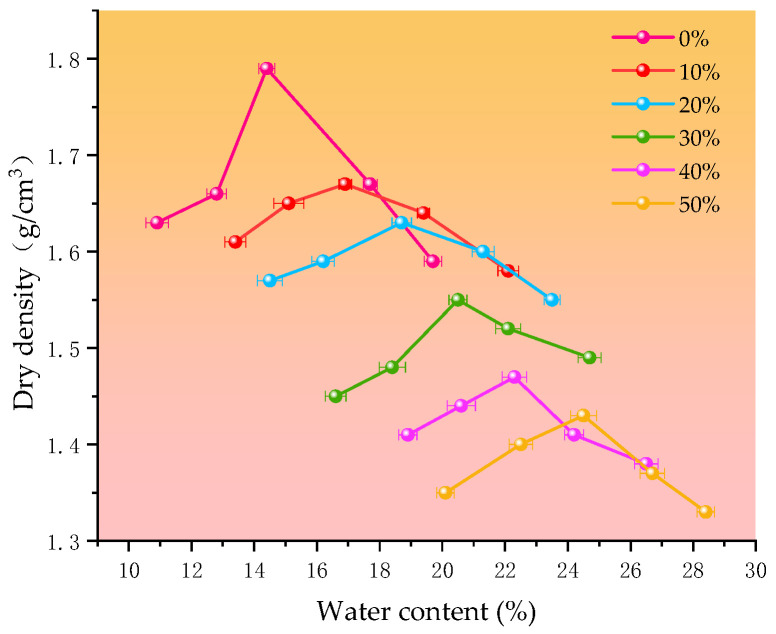
Moisture content and dry density of silt soil and different curing agent mixes.

**Figure 6 materials-18-00455-f006:**
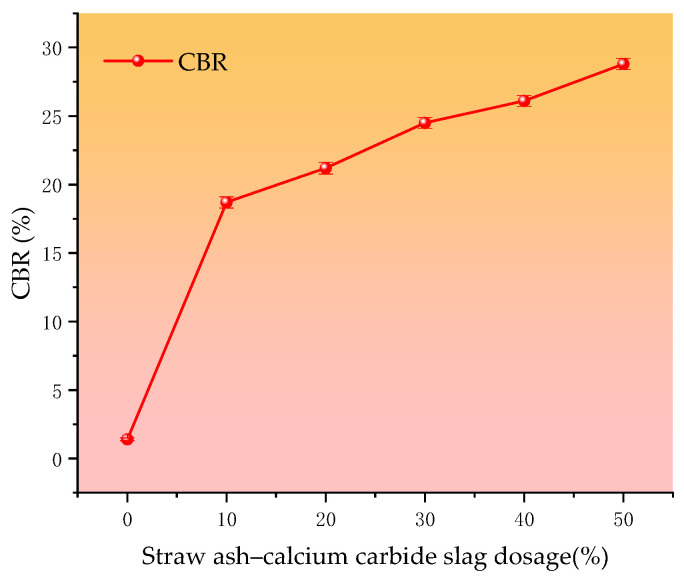
Relationship between Straw ash–calcium carbide slag dosage and CBR.

**Figure 7 materials-18-00455-f007:**
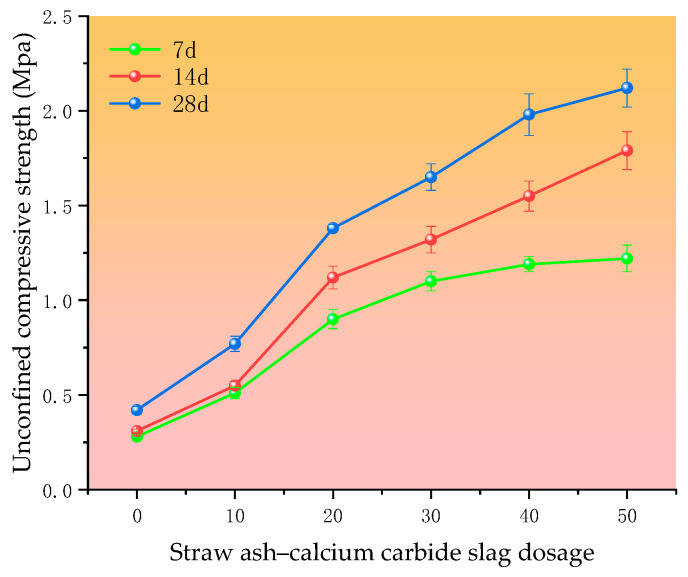
UCS at different curing cycles with different curing agent dosage.

**Figure 8 materials-18-00455-f008:**
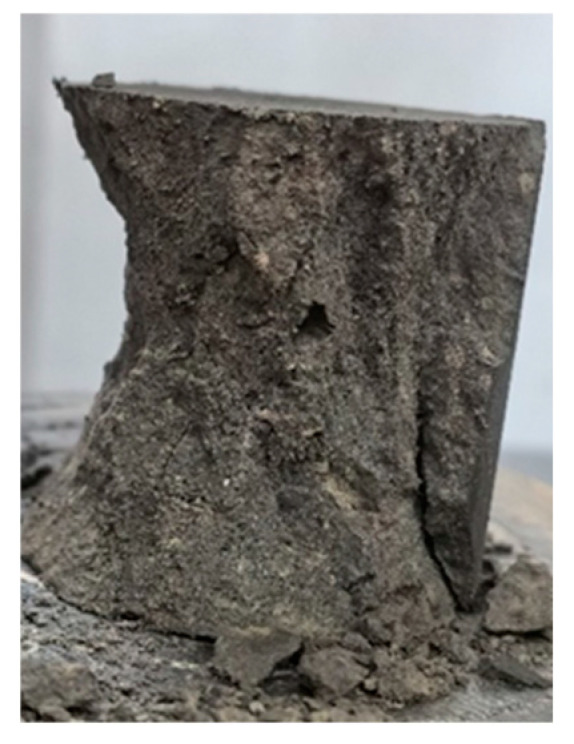
Kernel of solidated soil after compression damage.

**Figure 9 materials-18-00455-f009:**
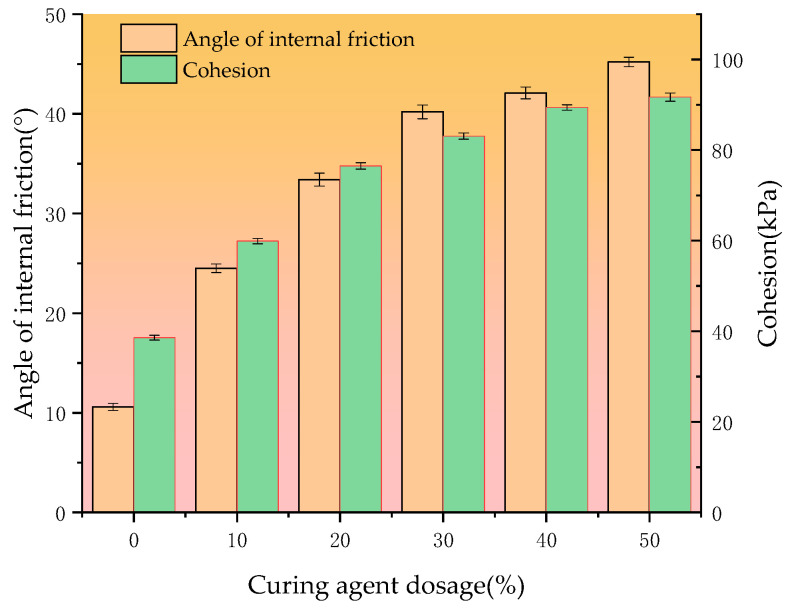
Relationship between shear strength and curing agent dosage.

**Figure 10 materials-18-00455-f010:**
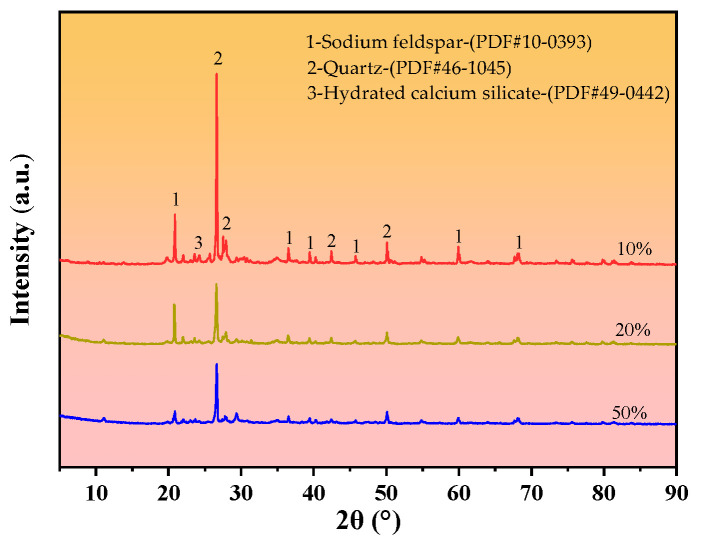
XRD pattern of cured silt soil.

**Figure 11 materials-18-00455-f011:**
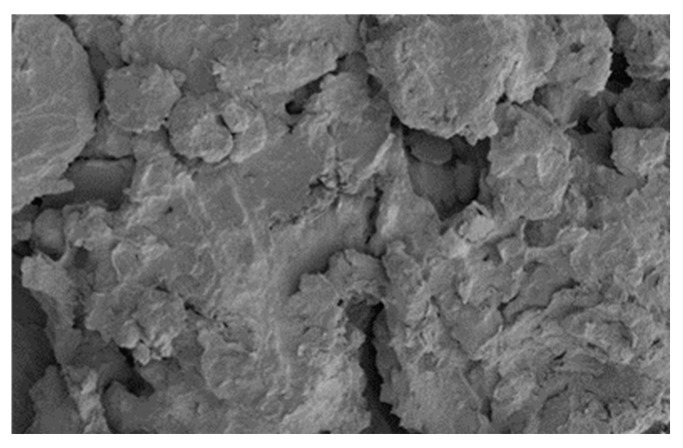
Silt soil without curing agent.

**Figure 12 materials-18-00455-f012:**
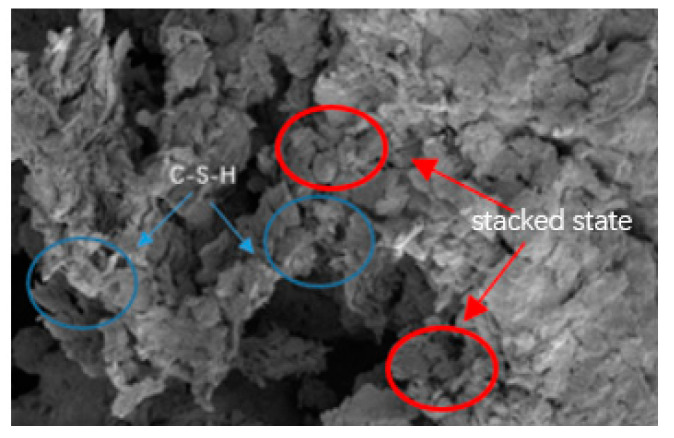
Cured soil with 20% curing agent.

**Table 1 materials-18-00455-t001:** Basic physical properties of silt soils.

Natural Moisture Content/%	Liquid Limit/%	Plastic limit/%	Plasticity Index	Liquid Index	Maximum Dry Density/ (g·cm^−3^)	Optimum Moisture Content/%	Specific Gravity
50.3 ± 1.1	38 ± 0.9	21.2 ± 0.6	16.8 ± 1.5	1.73 ± 0.13	1.79 ± 0.1	14.4 ± 0.4	2.56 ± 0.01

**Table 2 materials-18-00455-t002:** Main chemical composition content of raw materials (%).

Raw Material	Silt Soil	Straw Ash	Calcium Carbide Slag
SiO_2_	59.49	47.50	2.20
Al_2_O_3_	16.71	10.36	1.36
K_2_O	5.85	7.96	0.05
CaO	2.41	13.28	94.48
Fe_2_O_3_	9.84	6.60	0.83
MgO	1.82	2.60	0.10
Na_2_O	1.97	1.66	0.07

**Table 3 materials-18-00455-t003:** Minimum CBR requirements for roadbed fill.

Basement Area	Depth Below Road Surface	Minimum CBR (%)
Bed on the road	0–0.3	8

Since all the minimum bearing ratios sought for other roadbed areas are lower than those required for the upper roadbed, only the minimum CBR values for the upper roadbed are listed in the table.

**Table 4 materials-18-00455-t004:** Specimen morphology before and after UCS test.

Dosage of Curing Agent	Pre-Destruction	Post-Disruption
0%	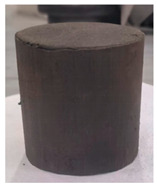	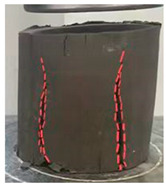
20%	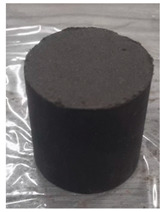	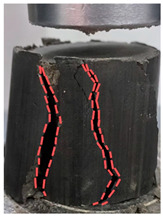

## Data Availability

The raw data supporting the conclusions of this article will be made available by the authors on request.

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
