# Peer review of "Mechanical Strength and Mechanism Analysis of Silt Soil Cured by Straw Ash–Calcium Carbide Slag"

_materials, 2025, doi:10.3390/ma18020455_

Round 1

Reviewer 1 Report

Comments and Suggestions for Authors

From the reviewer’s point of view, this text could become an interesting article about an important issue. However, this article is not ready for publication because its confusion in general, and its lack of clarity regarding its scope, some research decisions definition, research gap and structure.

1- The introduction lacks properly defining the scope or case study. Lines 37-39 are too vague. The definition of the scope needs further clarity and justification. This first section also lacks further explaining and justifying the selection of the two curing/stabilizing agents. The introduction also requires a more rigorous definition and justification of the gap that this project is covering. Lines 73-75 define a weak and vague gap. Lines 76-78 strongly state a clear gap but there is no justification, it lacks giving references to prove it. Finally, lines 78-85 lack proper location. Being a summary of the main findings need to be relocated in the abstract or in the conclusions, relying on the results and discussion. In its place the article lacks a brief description of the article main sections and contents.

2- Section 2 is rather confusing. The reviewer advices the authors to better present and organize its contents. For example, by adding a general explanation in the beginning presenting all the materials and tests before explaining each one in detail. Separating the materials and the experimental campaign in two main subsections would also help its understanding.

3- Section 3 is presented as the analysis of the mechanical properties. This title lacks indicating that this section also presents the main part of the results.

4- Considering the lacks in sections 2 and 3 a clearer structure of the paper is needed. The reviewer recommends the authors to use a more conventional structure of introduction, materials and methods, results and discussion and conclusions. The article also lacks a discussion that compares the analyzed results to former related research projects.

5- The introductory paragraph of the conclusions lacks a general explanation of the main contribution and findings of this project to its field of knowledge. Conclusion 1 lacks a clear explanation of its implications for the scope and case study.

6- Rethink the keywords so they are no words repeated in the title in order to multiply the text diffusion potential. Reorganize them, from general to particular.

7- The use of abbreviations has room for improvement in order to enhance the understanding of the article by potential readers. It is required to solve it by adding a list of abbreviations as well as repeating the meaning of the abbreviation the first time it appears in crucial sections as the conclusions.

8- Revise typewriting, p.e. “Figure19”, line 260 and use of English, same line “the Figure19”, “the Figure 22” line 367, Figure24 line 367, etc.

9- Revise the introduction of Figures in the manuscript before they appear, p.e. Fig 24, 25 and 26.

Reviewer 2 Report

Comments and Suggestions for Authors

This article describes the mechanical strength and mechanism analysis of silt soil cured by straw ash - calcium carbide slag.

To improve the manuscript, the authors should consider the following modifications:

(1) The authors either delete or place them in the Supporting Information (SI) section as these figures are not useful to the manuscript. These figures are Figure 4. Laser particle size meter; Figure 6. X-fluorescence spectrometer; Figure 7. Combined liquid-plastic limit tester; Figure 8. Experimental Procedure; Figure 9. Mixing the soil and then simmering the material overnight; Figure 10. Compacted drums after compaction; Figure 11. Curing in standard curing box; Figure 12. Electronic universal testing machine; Figure 13. Pavement Material Strength Tester; Figure 14. Expansion test by immersion in water; Figure 15. Strain Controlled Straight Shears; Figure 16. Scanning electron microscope; and Figure 17. X-ray diffractometer.

(2) This manuscript lacks an error analysis of the basic physical properties and mechanical properties of silt soils which are highly needed for readability purposes. The authors presented the basic physical properties of silt soils in Table 1. The authors should place the standard deviations of the Natural moisture content (%); Liquid limit (%); Plastic limit (%); Plasticity index; Liquid index; Maximum dry density (g·cm-3); Optimum moisture content (%); and Specific gravity in Tabe 1. The authors presented Figure 18. Trend of Boundary Moisture Content with Curing Agent Content; Figure 19. Moisture content and dry density of silt soil and different curing agent mix; Figure 20. Relationship between straw ash-electrolytic slag dosage and CBR; Figure 21. UCS at different curing cycles with different curing agent dosages; and Figure 23. Relationship between shear strength and curing agent dosage. The error bars of the Mechanical properties analysis such as Boundary moisture content; Solidification Property; California Bearing Ratio (CBR); Unconfined compressive strength; and Straight shear test are needed, which are presented in Figures 18 to 21 and 23 so that the reader will have an idea of the reproducibility of the data.

(3) The authors presented the XRD pattern of cured silt soil in Figure 24. The authors should assign all the diffraction peaks to the Miller indices lattice plane. What are the ICDD (International Center for Diffraction Data) card numbers for the XRD graph of the calcium carbide? It is suggested that the authors should include the ICDD pattern of calcium carbide in Figure 24 for a better understanding of the subject.

The submitted manuscript contains significant scientific insights, and the experimental data support the conclusions. However, the present submission requires major revisions before being considered for publication in the esteemed Materials in its current condition. I hope the authors will find my comments helpful.

Round 2

Reviewer 1 Report

Comments and Suggestions for Authors

From the reviewer’s point of view, this text could become an interesting article about an important issue. This version has improved following the reviewers’ comments. However, this article is not ready for publication because of the following.

1- The introduction still lacks properly defining the scope or case study. The definition is still too vague. The definition of the scope needs further clarity and justification. This first section also lacks further explaining and justifying the selection of the two curing/stabilizing agents. The introduction also requires a more rigorous definition and justification of the gap that this project is covering because it defines a weak and vague gap. The lines that strongly state a clear gap lack justification, this is lack giving references to prove it. Finally, the last lines of the introduction lack proper location. Being a summary of the main findings need to be relocated in the abstract or in the conclusions, relying on the results and discussion. In its place the article lacks a brief description of the article main sections and contents.

2- The use of abbreviations still has room for improvement in order to enhance the understanding of the article by potential readers. It is still required to solve it by adding a complete and proper list of abbreviations as well as repeating the meaning of the abbreviation the first time it appears in crucial sections as the conclusions.

Reviewer 2 Report

Comments and Suggestions for Authors

Dear Authors: Many thanks for your sincere efforts in improving your manuscript. The revised article is highly satisfactory and merits acceptance for publication in the Materials.

Author Response

Thank you very much for taking the time to review this manuscript.